# An Isolate of *Streptococcus mitis* Displayed In Vitro Antimicrobial Activity and Deleterious Effect in a Preclinical Model of Lung Infection

**DOI:** 10.3390/nu15020263

**Published:** 2023-01-04

**Authors:** Elliot Mathieu, Quentin Marquant, Florian Chain, Edwige Bouguyon, Vinciane Saint-Criq, Ronan Le-Goffic, Delphyne Descamps, Philippe Langella, Thomas A. Tompkins, Sylvie Binda, Muriel Thomas

**Affiliations:** 1Micalis Institute, Institut National de Recherche pour L’Agriculture, L’Alimentation et L’Environnement (INRAE), AgroParisTech, Université Paris-Saclay, UMR1319, F-78350 Jouy-en-Josas, France; 2Paris Center for Microbiome Medicine (PaCeMM) FHU, AP-HP, F-75571 Paris, France; 3Université Paris-Saclay, INRAE, UVSQ, VIM, F-78350 Jouy-en-Josas, France; 4Laboratoire VIM-Suresnes, Hôpital Foch, F-92150 Suresnes, France; 5Lallemand Bio Ingredients, Montreal, QC H4P 2R2, Canada; 6Lallemand Health Solutions, Montreal, QC H4P 2R2, Canada

**Keywords:** *Streptococcus*, inhibition, anti-microbial, *Pseudomonas aeruginosa*, *Staphylococcus aureus*, *Haemophilus influenzae NTHi*, infection

## Abstract

Microbiota studies have dramatically increased over these last two decades, and the repertoire of microorganisms with potential health benefits has been considerably enlarged. The development of next generation probiotics from new bacterial candidates is a long-term strategy that may be more efficient and rapid with discriminative in vitro tests. *Streptococcus* strains have received attention regarding their antimicrobial potential against pathogens of the upper and, more recently, the lower respiratory tracts. Pathogenic bacterial strains, such as non-typable *Haemophilus influenzae* (*NTHi*), *Pseudomonas aeruginosa* (*P. aeruginosa*) and *Staphylococcus aureus* (*S. aureus*), are commonly associated with acute and chronic respiratory diseases, and it could be interesting to fight against pathogens with probiotics. In this study, we show that a *Streptococcus mitis* (*S. mitis*) EM-371 strain, isolated from the buccal cavity of a human newborn and previously selected for promising anti-inflammatory effects, displayed in vitro antimicrobial activity against *NTHi*, *P. aeruginosa* or *S. aureus*. However, the anti-pathogenic in vitro activity was not sufficient to predict an efficient protective effect in a preclinical model. Two weeks of treatment with *S. mitis* EM-371 did not protect against, and even exacerbated, *NTHi* lung infection.

## 1. Introduction

Live microorganisms have been used safely for a long time in human food (e.g., fermented food) and, more recently, in formulations of probiotics. Most of the bacterial strains used as probiotics belong to a limited number of genera including mostly *Lactobacillus* and *Bifidobacterium* and, to a lesser extent, *Enterococcus*, *Escherichia coli* and *Streptococcus,* as well as the nonpathogenic yeast *Saccharomyces*. The probiotic strains provide their beneficial effect through different mechanisms of action that involve the restoration of microbial diversity, modulation of the immune system, production of regulatory metabolites and competitive exclusion of pathogens [1,2]. This last antimicrobial effect is mediated by the ability of the beneficial strains to compete for the nutrient sources, modulate the abiotic environment and produce bacteriocins or bacteriocin-like molecules [3]. Interest is growing regarding the capacity of new bacterial isolates from a human origin to prevent or treat bacterial infection. However, the development of probiotics from new bacterial candidates is a long-term strategy that requires in silico, in vitro and in vivo testing and, ultimately, clinical validation [4,5]. It is therefore helpful to use efficient and discriminative in vitro screening to highlight the most promising candidates in order to reduce the in vivo experimentation.

Acute and chronic respiratory diseases are among the most common causes of severe illness and death worldwide [6]. It has been estimated that nearly one billion people suffer from respiratory diseases, and the burden is increasing every year [6]. Microbial infection is a hallmark of respiratory diseases. Thus, pathogens such as *Haemophilus influenzae* (*H.i*), *Pseudomonas aeruginosa* (*P. aeruginosa*) and *Staphylococcus aureus* (*S. aureus*) are common colonizers in cystic fibrosis (CF), asthma, chronic obstructive pulmonary disease (COPD) and bronchiectasis [7,8,9,10,11,12,13]. The persistent chronic bacterial infections contribute to the exacerbation of tissue damage and the decline in pulmonary functions [10,14]. *H.i*, *P. aeruginosa* and *S. aureus* are also commonly isolated from patients suffering from community-, hospital- and ventilator-acquired pneumonia [13,15,16,17,18,19]. The effects are exacerbated, as conventional antibiotic treatments fail due to massive antibiotic resistance. Therefore, the development of new alternative therapeutic interventions is of major importance.

Members of the *Streptococcus* genera, such as *Streptococcus mitis* and *salivarius*, are recognized as commensals in various microbiota of the human body. They are the first inhabitants of the oral cavity and favour its colonization by other members of the early-life buccal microbiota [20,21]. Moreover, *Streptococcus* species are common members of the lung microbiota [22,23,24]. The current understanding of the lung microbiota characteristics suggests its neutral dispersion from the buccal cavity [24,25,26]. The buccal cavity is a reservoir of microorganisms that migrate to the lungs and may display probiotic potential for respiratory health. Previous in vitro studies have evaluated the inhibiting properties of commercially available *Streptococcus salivarius* (*S. salivarius*) strains on diverse pathogens of the upper respiratory tract [27]. For example, the *S. salivarius* strain K12 reduces pneumococcal adherence to pharyngeal epithelial cells and inhibits mycelial growth of *Candida albicans* [27,28]. In addition, intranasal instillation of *Streptococcus mitis* (*S. mitis*) promotes protection against pneumococcal lung infection [29]. *Streptococcus* strains isolated from the buccal cavity of young children may protect against respiratory pathogens.

We have previously isolated an *S. mitis* strain (*S. mitis* EM-371) from the buccal cavity of a human newborn [30]. The objectives of this study were to (i) investigate the EM-371 strain genome in silico to identify genes potentially conferring antibacterial properties; (ii) validate, in vitro, the growth inhibitory effects of *S. mitis* EM-371 against multiple strains of *H. influenzae*, *S. aureus* and *P. aeruginosa* species; and (iii) establish whether this strain protects against bacterial infection in vivo in a murine model and, thus, if in vitro testing could predict the in vivo response. In this study, we identified multiple genes with potential importance for primo-colonisation and antibacterial effects. Together with our previously published results showing that *S. mitis* EM371 has no pro-inflammatory properties, this shows that this strain harbours several beneficial properties that may be promising for future probiotic development. Moreover, agar-well diffusion assays demonstrated an inhibiting activity of *S. mitis* EM371 against type strains and clinical isolates of *H. influenzae*, *S. aureus* and *P. aeruginosa.* However, in vivo tests have shown that two weeks of preventive treatment with *S. mitis* EM371 did not confer any protection against *NTHi* lung infection in mice when considering weight loss, lung bacterial burden, immune infiltration, histopathology and function (Penh index). Our study shows that an anti-pathogenic in vitro activity is not necessarily a clue to an efficient protective effect in a preclinical model. Both adequate and physiologically relevant in vitro and in vivo models are thus essential for the efficient characterisation and screening of bacterial strains for the development of future probiotics.

## 2. Materials and Methods

### 2.1. Bacterial Strains, Media, Growth Conditions

The *S. mitis* EM-371 was previously isolated from the oral cavity of a 5-day-old human newborn (as described in [30]). The strain was cultivated in a brain heart infusion (BHi) (Ref #237500, BD/Difco, Fisher Scientific, Thermo Fisher Scientific, Waltham, MA, USA) broth medium supplemented with 5 g·L^−1^ of yeast extract (Ref #212750, BD/Difco, Fisher Scientific, Thermo Fisher Scientific, Waltham, MA, USA). The strains of *NTHi* 86-028NP, *S. aureus* Chicago and *P. aeruginosa* CHA, 16792a and 208156470c were generous gifts from Dr Lhousseine Touqui (INSERM UMR938; Hôpital Saint-Antoine, Paris, France). The *S. aureus* strains ATCC 33591, ATCC 43300, ATCC 27217, ATCC 29213 were kindly provided by Dr Romain Briandet (INRAE UMR1319, Institut MICALIS, Jouy-en-Josas, France). Finally, the *H. influenzae* strains ATCC 33391 and 162 and the *S. aureus* CNRZ 587 and Chicago were part of our collection. The *P. aeruginosa* and *S. aureus* were cultivated in the BHi broth medium supplemented with 5 g·L^−1^ of yeast extract. Hemin (10 µg/mL) and b-Nicotinamide adenine dinucleotide (NAD, 10 µg/mL) (Ref #N6622-250MG, Sigma-Aldrich, St. Louis, MO, USA) were added to the BHi broth to cultivate the *H. influenzae* strains. The cultures were grown at 37 °C and passed once before the day of the experiment.

### 2.2. gDNA Extraction and Genome Analysis of S. mitis EM-371

Genomic DNA (gDNA) from *S. mitis* EM-371 was extracted using phenol-chloroform. The *S. mitis* EM-371 was thawed, cultivated overnight in BHi, and the bacterial culture was divided into aliquots before centrifugation. The supernatant was removed, and freshly prepared lysis buffer was added to the pellet. After one-hour incubation at 37 °C, Proteinase K and AL buffer (from DNeasy Blood and Tissue Kit, Ref #69504, Qiagen, Hilden, Germany) were added, and the tubes were incubated in a dry bath at 56 °C for 30 min. The bacterial lysates were transferred to PhaseLock tubes (Ref #2302830, Quantabio, Beverly, MA, USA), and RNase A (Cat #11119915001, Sigma, St. Louis, MO, USA) was added; the tubes were then incubated at 37 °C for 30 min. The first phenol extraction was realized by adding phenol:chloroform:isoamyl alcohol (25:24:1). After centrifugation, the supernatant was collected, transferred to a new PhaseLock tube, and a second phenol extraction was performed. Once again, the supernatant was collected, isopropanol was added to each tube and DNA was allowed to precipitate. Following the precipitation, the pellets were washed twice with cold 70% ethanol and dried in a flow hood for about 20 min. The pellets were resuspended with RNAse/DNAse-free water and gDNA concentrations, and the purity was determined using the Nanodrop. A 260/280 ratio of 1.8 to 2.0 was considered high-purity gDNA. To ensure a good quality of gDNA, the samples were run on a 0.8% agarose gel. A single band running above the DNA ladder indicates a good gDNA quality. In contrast, smearing indicates bad quality. A minimum of 10 µg of good quality gDNA was sent to the Yale Center for Genomic Analysis (YCGA, New Haven, CT, USA) for genome sequencing with the Pacific Biosciences (PacBio) long-reads technology using a 3–10 Kb library. 

The genome analysis was performed using the PathoSystems Resource Integration Center (PATRIC, Version 3.5.31, https://www.bv-brc.org/) and Rapid Annotation using Subsystem Technology (RAST, Version 2.0, https://rast.nmpdr.org/). In PATRIC, the services Assembly, Annotation, Comprehensive Genome Analysis and BLAST were used. In RAST, we used the tools Subsystem Feature Count, Function-Based Comparison and Sequence-Based Comparison.

### 2.3. Agar Well Diffusion Method

The agar well diffusion test was performed using the BHi broth medium and BHihyea agar (BHi, hemin, yeast extract, agar). First, a layer of 10 mL BHihyea agar was poured into a 9 cm Petri dish and allowed to cool down. The first agar layer served as the substrate for the *Streptococcus* EM-371. To prepare the second layer, a mix of BHi/BHihyea (50/50, *v*/*v*) (BHim) was made and kept melted at 60 °C until needed. A required volume of 24 h pathogen culture was added to the BHim (cooled down to 55 °C) to reach a final dilution of 1/100. Then, 20 mL of the pathogen/BHim mix was poured onto the first agar layer and allowed to cool down. Once the medium had solidified, three wells, 7 mm each, were cut onto the top agar layer only (pathogen/BHim). Following that, 20 µL of either the medium control, antibiotic control (+) or *Streptococcus* EM-371 (S) were placed into each well. The plates were incubated at 37 °C and the growth inhibition zones were measured 24 h later. The antibiotic controls used in this study were colistin (120 ng/well) against the *P. aeruginosa* strains and chloramphenicol (20 µg/well) against the *H. influenzae* and *S. aureus* strains. Note that all the experiments with the *H. influenzae* strains were performed using the BHi medium supplemented with hemin and nicotinamide-adenine-dinucleotide (NAD). Moreover, the *NTHi* was spread onto the second layer instead of the first layer. Overall, the strains were tested at least twice.

### 2.4. Animals and Procedures

The animal experiments were approved by the Ministère de l’Enseignement supérieur de la Recherche and the local ethics committee under the registration number APAFIS# 2018092818468900_V4. Six-weeks-old specific-pathogen-free (SPF) BALB/cJRj male mice, purchased from Janvier (Le Genest, St Isle, France), were bred and housed under FELASA pathogenic conditions in our animal facilities (IERP, INRAE, Jouy-en-Josas, France). A total of 32 mice were used for the experiment. The mice were randomized into four groups of eight mice. Briefly, two groups of mice received, intranasally, (i.n) 10 µL of the *Streptococcus* EM-371 strain (1.10^6^ CFU/10 µL) every two days for two weeks, while the other two groups received the vehicle (PBS). At day 14, the mice were anesthetized using ketamine and xylazine by intraperitoneal injection (0.1% ketamine + 0.06% xylazine, 150 µL/20 g of mouse weight) and infected i.n with 1.10^8^ CFU of *NTHi* 162 in a volume of 50 µL in 5% BHi in PBS or received the vehicle (5% BHi in PBS). BHi was used to maximise the viability of *NTHi*, as previously described by Siggins and colleagues [31]. The animals were weighed 24 h and 48 h after infection and at day 16; lung function was assessed by plethysmography before the mice were culled by intraperitoneal injection of a lethal dose of ketamine and xylazine (300 mg ketamine + 36 mg xylazine per kg of mice weight), and their organs were sampled (see below).

### 2.5. Plethysmography

The pulmonary functions were assessed 48 h post-infection using whole-body plethysmography on conscious unrestrained mice. The animals were individually introduced to a plethysmography chamber (emka Technologies, Falls Church, VA, USA) and 10 respiratory cycles were recorded for each animal. The pulmonary functions of all the mice were recorded in the same chamber.

### 2.6. Sample Collection

After culling, the right bronchus was clamped and bronchoalveolar lavage (BAL) was performed on the left lobes with PBS. A portion of the BAL was serially diluted and plated onto BHihyea/NAD/hemin to quantify the levels of *NTHi* 162. After 48 h of incubation at 37 °C, the colonies were enumerated. The remaining portion of the BAL was centrifuged (700× *g*, 4 °C, 5 min), and the BAL cells were cytocentrifuged (Cytospin 5, 1000 rpm; RT, 10 min) on microscope slides (Superfrost, Thermo Scientific, Braunschweig, Germany), and then stained with May-Grünwald and Giemsa. The left lobe was then fixed with 4% paraformaldehyde and further embedded in paraffin for histological examination. The right lobe was unclamped and used for flow cytometry.

### 2.7. Histology

Five-micrometre lung sections were stained with hematoxylin/eosin/saffron and captured using PanoramicViewer software (3DHISTECH, Budapest, Hungary).

### 2.8. Flow Cytometry

The cell suspensions were prepared from the lung tissue. Briefly, the lung tissue was homogenized using a Precellys (10 s, 5000 rpm) Bertin Instruments, Rockville, MD, USA), and the cell suspensions were filtered using 40 µm cell strainers (Ref # CLS431750-50EA Corning, Sigma-Aldrich, St. Louis, MO, USA). After centrifugation, the cells were distributed in a 96-well plate and stained in PBS for 20 min at 4 °C with Zombie Aqua (BV510, BioLegend, San Diego, CA, United States), which was used to determine live/dead cells. After incubation, the cells were washed and centrifuged (1800 rpm, 10 min, 4 °C). The cells were then stained in FACS buffer (D-PBS + 2% FBS + 2 mM EDTA) with Alexa Fluor-conjugated anti-CD45.2 (Sony Biotechnology Europe, Weybridge, UK), PerCP-Cy5.5-conjugated anti-CD3e (BioLegend), APC-conjugated anti-CD19 (clone 1D3) (BD), Brilliant Violet-conjugated anti-Ly6C (clone HK1.4) (BioLegend), anti-CD11b (clone M1/70) (Sony) and anti-Siglec-F (clone DIH9) (BD), PE-conjugated anti-Ly6G (clone 1A8) (BD), PE-Cy7-conjugated anti-CD11c (clone N418) (Sony) and, finally, APC-Cy7-conjugated anti-IA/IE (clone M5/114.15.2) (BioLegend). The Fc receptors were blocked using Human TruStain FcX (anti-CD16/CD32) (Sony). The data were collected using the FACS Fortessa (BD Biosciences, Franklin Lakes, NJ, USA) and analysed using the FlowJo version 10 (Treestar, Ashland, OR, USA). The cell doublets were excluded using an FSC-H versus FSC-A plot. The gating strategy can be found in Appendix A.

### 2.9. SCFA Analysis

The SCFA (acetate, propionate and butyrate) levels in the culture supernatants were determined by gas chromatography (Nelson 1020, Perkin Elmer, St Quentin-en-Yvelines, France). The overnight culture was centrifuged, and the supernatant was collected. The proteins were precipitated using a saturated phosphotungstic acid solution. A volume of 0.1 mL of the supernatant was analysed using a gas–liquid chromatograph (Autosystem XL; Perkin Elmer, Saint-Quentin-en-Yvelines, France). All the samples were analysed in duplicate. The data were collected, and the peaks were integrated using Turbochromv6 software (Perkin Elmer, Courtaboeuf, France).

### 2.10. Statistical Analysis

Non-parametric Mann–Whitney or one-way ANOVA Bonferroni’s multiple comparison tests were used to compare unpaired values. To compare the two groups, Unpaired t-tests were performed. To compare the growth curves, the areas under the curve were calculated, and data were used to perform a one-way ANOVA (GraphPadPrism software, GraphPad Software, Inc., San Diego, CA, USA). Significance is represented as: *: *p* < 0.03, **: *p* < 0.002, ***: *p* < 0.0002 and ****: *p* < 0.0001.

## 3. Results

### 3.1. Probiotic Potential of S. mitis EM-371

We have previously isolated bacterial strains from the buccal cavity of human newborns and showed that primo-colonizing bacteria impact the immunity and morphology of the lung epithelial cells, with specific effects depending on the phylogeny of the strains [30]. From these primo-colonizing strains, we have identified an *S. mitis* (*S. mitis* EM-371) strain that displays interesting promising probiotic capacities (Table 1). A genome analysis, using RAST and PATRIC databases (both are microbial genome annotation and sequence analysis tools), revealed the presence of sequences coding for several proteins related to adhesion. In particular, we identified the wall anchor and surface proteins CbpD, lytC, pce, pavA, GAPDH, lmb, NanA, Eno, ZmpB and PsaA in the *S. mitis* EM371. We also identified other genes, blpT, blpR, blpH, pncG, blpL, blpZ, pncP, tsaD, cibA, comA, comB, comD and comE, which are associated with the production of bacteriocins. Furthermore, the *S. mitis* EM-371 modulated cytokine/chemokine production by the bronchial epithelial cells. This strain modulated the release of some cytokines, chemokines and growth factors (e.g., IL-7, MIF, basic FGF, CCL1, CCL21, CCL25 and CCL26) but did not display a pro-inflammatory cytokine profile [30]. In addition, in the bacterial culture supernatant, the *S. mitis* EM-371 produced up to 4.1 mM of acetate, which may modulate the immune system [32]. 

### 3.2. S. mitis EM-371 Inhibits the Growth of Bacterial Respiratory Pathogens In Vitro on Agar-Diffusion Test

The ability of *S. mitis* EM-371 to inhibit the growth of bacterial respiratory pathogens was assessed using an agar well diffusion assay. The agar well diffusion method is commonly used to evaluate the antimicrobial activity of microbial compounds or microbial agents. Figure 1 shows the growth inhibition zone (i.e., the zone around the well where the bacteria are not growing) induced by *S. mitis* EM-371 (in Figure 1: S = *S. mitis* EM-371; + = antibiotics control). *S. mitis* EM-371 induced an inhibition >1 cm against the type strain of *H. influenzae* ATCC 33391 and against two clinical isolates *NTHi* 86-028NP and *NTHi* 162 (Figure 1a). Moreover, we showed that the *S. mitis* EM-371 inhibited the growth of different *S. aureus* strains (Figure 1b). It inhibited both methicillin-resistant *S. aureus* (MRSA) (ATCC 33591 and ATCC 43300) and non-MRSA (ATCC 27217 and ATCC 29213) to the same extent. The *S. mitis* EM-371 induced a larger growth inhibition zone than the antibiotics control against *S. aureus* ATCC 33591 (Figure 1b, growth inhibition diameter: 1.4 and 1.05 cm, respectively for S and +). Last, we assessed the inhibiting capacities of the oral commensal against 3 strains of *P. aeruginosa* (Figure 1c). The smallest growth inhibition zone was observed against *P. aeruginosa* CHA (Figure 1c, growth inhibition diameter: 0.86 cm). Interestingly, no inhibition was observed with the supernatant of the *S. mitis* EM-371 culture against all three pathogens (data not shown). The *S. mitis* EM-371 displayed inhibiting properties against Gram-positive (*S. aureus*) and Gram-negative (*P. aeruginosa* and *H. influenzae*) pathogens, with a particular effect against *NTHi*.

### 3.3. Preventive Treatment with S. mitis EM-371 Did Not Protect the Mice against Haemophilus influenzae (NTHi) Lung Infection in Mice

We then tested the potential protective effect of the *S. mitis* EM-371 against lung infection triggered by *NTHi* in mice. Six-week-old BALB/cJRj mice received two weeks of preventive treatments with *S. mitis* EM-371 (Figure 2a, 1.10^6^ CFU/10 µL intranasally every 2 days). On day 14, the mice received the NTHi at 1.10^8^ CFU/50 µL intranasally. On day 16, broncho-alveolar lavage (BAL) and lung tissue samples were collected. The mice that were infected with the *NTHi* (on day 14) lost about 7% of their weight on the first day and up to 13% 48 h post-infection (Figure 2b). The weight loss was enhanced by the *S. mitis* strain, with a mean of 17% loss of body weight in 2 days (Figure 2b, not significant). 

At 48 h after infection, the *NTHi* bacterial loads in the BAL were 1.17.10^4^ CFU/mL, both in the mice that received the *NTHi* or the *S. mitis* EM-371 + *NTHi* (Figure 3a). The total cell number in the BAL was significantly higher for the *NTHi* + EM-371 group when compared to the other conditions (Figure 3b). We observed that the two bacteria combined have a synergistic effect on cell infiltration. Indeed, the treatment with the *S. mitis* EM-371 alone increased the level of infiltrating cells in the same manner as the *NTHi* (Figure 3b). We also found that the level of neutrophils in the BAL was significantly increased in response to the treatment of the *NTHi* + EM-371 (Figure 3c). However, there was no significant difference in the levels of macrophages in the BAL between the different conditions (Figure 3d). 

The cellular characterization of the lung tissue by flow cytometry also permitted us to highlight an increased frequency of the neutrophils (Ly6Ghigh) in the lungs (Figure 4a). The lung infection with the NTHi increased the level of conventional dendritic cells (cDC) CD11b+ (Figure 4b). The same frequency of cDC CD11b+ was observed in the lung tissue of mice treated with *S. mitis* EM-371. However, preventive treatment with *S. mitis* EM-371 restored the proportion of cDC CD11b+ induced by the *NTHi* (Figure 4b). We observed that the proportion of alveolar macrophages (AM) decreased in favour of the interstitial macrophages (IM) and monocytes (Mo) (Ly6C-) but also in favour of the neutrophils (Figure 4). 

*NTHi* considerably reduced the presence of AM in lung, and the treatment with *S. mitis* EM-371 did not prevent this drop. The innate response is taking place and the influx of neutrophils and IM appear to be deleterious for the lungs, as shown in the panel of lung sections presented in Figure 5. We observed that the *NTHi*-challenged mice showed an increased level of inflammatory cell recruitment near the airway (Figure 5a). The epithelium thickness was slightly increased, but not significantly, in the mice infected with *NTHi* (Figure 5b). When administered alone, the *Streptococcus* induced similar levels of inflammatory cell recruitment and bronchus epithelium thickening to the PBS. However, the lungs of mice that received both the *S. mitis* EM-371 and *NTHi* showed a significant increase in the thickness of the epithelium when compared to the PBS group (Figure 5b, PBS vs *NTHi* + 371, *p*-value < 0.0002). Before the autopsy, we analysed the lung functions of the mice using whole-body plethysmography. As shown in Figure 5c, the mice infected with *NTHi* had strong respiratory distress (Penh > 1), and this was exacerbated by the *S. mitis* EM-371 treatment. Instead of being protective, the pre-treatment with *Streptococcus* worsened the disease outcomes driven by *NTHi*. 

## 4. Discussion

The objectives of the present study were to test, in vitro and in vivo, a potential probiotic *S. mitis* strain on the inhibition of common respiratory pathogens and to challenge the relevance of this strategy in a preclinical model of an *NTHi* lung infection. Although giving important indications on the ability of the *S. mitis* isolate to inhibit, in vitro, the growth of type strains and clinical isolates of *H. influenzae*, *P. aeruginosa* and *S. aureus*, we did not show any protection in a model of lung infection by *NTHi* in mice. Even worse, prophylactic treatment with *S. mitis* EM-371 aggravated the disease outcomes. 

Microbiota studies have dramatically increased in the two last decades, thanks to the development of adapted culturing methods, high-throughput genome sequencing strategies and tools to analyse large amounts of genomic data. Therefore, the repertoire of microorganisms with potential health benefits has considerably expanded [33]. For this study, we identified a promising, beneficial *S. mitis* strain. An initial, interesting characteristic of this strain is that it has been isolated from the buccal cavity of a human newborn. *S. mitis* is a pioneer member of the buccal cavity and plays a fundamental role in the maintenance of oral homeostasis [20,21]. *S. mitis* expresses several adhesins that may promote adherence to host tissue [34,35]. The genome analysis of the *S. mitis* EM-371 revealed the presence of numerous genes associated with choline-binding protein, laminin-binding protein, fibronectin-binding protein, plasminogen-binding and other surface proteins that may be putative adhesins. Although there is no consensus, adherence is sometimes a criterion for the selection of potential probiotics [4,5,36,37]. Another recognised mechanism of action for probiotics is an antimicrobial action against pathogens by producing antimicrobial molecules such as bacteriocins. We have identified, in the sequence of *S. mitis* EM-371, bacteriocin-associated encoding genes in the blp loci (blpT, blpR, blpH, pncG, blpL, blpZ and the CAAX protease pncP), cibAB (tsaD, cibA) and the locus upstream comAB (comA, comB, comD, comE). *S. mitis* EM-371 does not possess most of the usual *Streptococcus* bacteriocins [38], salA, sboB, sibA and scnA, but harbours the srtA gene. The comparison of the genome of *S. mitis* with two closely related strains, *S. pneumoniae* and *S. pseudopneumonia*, showed it contains a smaller repertoire of genes associated with bacteriocin production [34,39]. The genetic basis of bacteriocin production for *S. mitis* is not fully understood, but some members of the species have shown interesting inhibiting properties [29,39]. Note that the objective of the present work was not to reconstruct the gene clusters involved in bacteriocin production but rather to get clues on the potential antimicrobial properties of our strain. In a previous study, we showed that *S. mitis* EM-371 modulates gene expression and cytokine/chemokine/growth factor production in bronchial epithelial cells [30]. The strain did not display a pro-Th1 inflammatory profile but was able to modulate inflammatory pathways and secretion of immune-related proteins in BEAS-2B cells. The regulation of host immune response is an important mechanism of action of probiotics that can also be mediated by metabolites such as SCFA. Several studies have highlighted the impact of SCFA derived from the intestinal compartment on the regulation of immune cells involved in host response to lung infection [32,40,41,42]. In culture, *S. mitis* EM-371 was able to produce acetate and negligible amounts of propionate and butyrate. Altogether, these arguments sustain the potential promising development of *S. mitis* EM-371 as a probiotic.

We were particularly interested in the antimicrobial activity of *S. mitis* EM-371. Here, we tested the capacity of an oral *S. mitis* isolate to inhibit the growth of several strains of respiratory pathogens. Remarkably, the *S. mitis* EM-371 strain showed inhibiting properties against both Gram-positive (*S. aureus*) and Gram-negative (*P. aeruginosa* and *H. influenzae*) pathogens, with a particular effect against *NTHi* (Figure 1a). However, we showed that direct contact with the pathogens was required for the *S. mitis* EM-371 to perform its antimicrobial activity, and the culture supernatant of *S. mitis* EM-371 did not induce a growth inhibition zone. Interestingly, a study by Ikryannikova et al. showed that an *S. mitis* isolate was able to inhibit the growth of *Moraxella catarrhalis* strains in vitro, and this ability was kept even after killing the strain [39]. The basis of inhibition of *S. mitis* remains unclear, but the results of the in vitro experiments opened promising perspectives and showed that *S. mitis* EM-371 could be a potential beneficial strain, particularly in the management of respiratory infections. We, therefore, evaluated the beneficial effect of *S. mitis* EM-371 in a preclinical model of lung infection and, ultimately, validated the relevance of the in vitro method.

The preclinical model of an *NTHi* lung infection has been previously described by Siggins and colleagues [31]. Examination of the mice 24 and 48 h after the infection showed a reduced activity of the individuals that received both treatments compared to those that only received the pathogen. The loss of activity reflected a significant weight loss and diminished pulmonary function. The *NTHi*-challenged mice showed increased levels of inflammatory cells recruited near the airway and epithelium thickening when compared to mice that were not infected. We also observed that preventive treatment with *S. mitis* EM-371 before the *NTHi* infection resulted in a significant increase in inflammatory cell recruitment and epithelium thickening. Lung function was clearly impacted by the *NTHi* as the mice showed strong respiratory distress. This parameter was significantly increased in the mice that received the *NTHi* + EM-371, indicating that the mice had more difficulties in breathing when pre-treated with *S. mitis*. In humans, persistent infection by *NTHi* in the lungs of COPD patients accelerates chronic airway inflammation and loss of pulmonary function [43]. An immune response to *NTHi* involves numerous cells and mediators that participate together to clear the pathogen from the lungs. As the first line of defence, macrophages and neutrophils are recruited to the airways to clear the invading pathogens [16,43]. As observed in previous studies, the *NTHi* induced prominent airway neutrophilia [31,44]. Interestingly, we observed an increased level of neutrophils in the BAL from the mice that received *S. mitis* EM-371 prior to infection. This increase might be due to the fact that neutrophils are acting against both the *S. mitis* and *NTHi*. Lung-resident neutrophils were recruited at the same frequency for both groups that received the *NTHi,* with or without preventive treatment. Herein, no significant differences were observed for the level of macrophages in the BAL but, in the lung, all conditions reduced significantly the frequency of alveolar macrophages. These cells patrol the lungs and are able to recognize and phagocyte foreign elements [45]. Thus, we would have expected an increased level of alveolar macrophages in response to the *NTHi* [44]. However, infection of the mice resulted in an increase in the frequency of the cell population IM + Ly6C-Mo, which was partially restored with the use of the preventive treatment. The increased presence of neutrophils in the lungs of the mice that received the preventive treatment did not favour the clearance of the *NTHi,* as observed with the number of viable cells recovered from the lungs. Several other immune cells, such as cDC and lymphocytes, are recruited to the lung in response to lung infection by *NTHi* [16,44]. As major interfaces between innate and adaptive immunity, the cDC cells are able, via a large repertoire of receptors, to recognize pathogens and initiate an acute inflammatory response [46]. Upon stimulation, the cDC cells produce a wide range of cytokines and are able to drive CD4+ and CD8+ T cell response through antigen presentation via the MHC-II molecule [47]. In our study, the two weeks of treatment with *Streptococcus* significantly increased the frequency of DC CD11b+ (cDC2) but not of DC CD11b- (cDC1). The increased frequency of the cDC subset caused by our preventive treatment was not linked to an increased frequency of T cells. When addressing the frequency of B and T cells in the lung tissue, we observed that the T lymphocytes were negatively regulated in the infected mice when compared to the PBS control (Appendix A). Altogether, our animal data show that the pre-treatment with *S. mitis* EM-371 aggravated the disease outcomes. In addition, in the context of moderate exposure to *NTHi*, the pre-treatment with *S. mitis* EM-371 exacerbated the disease outcome. We have preliminary data showing that a cocktail of two other bacterial strains showed a neutral effect on the model (data not shown), suggesting that the modulation of the infection is strain-specific. No protection and no aggravation of the infection were observed. The mechanisms underlying the exacerbated effect of *S. mitis* were not deciphered in this study and remain unknown. However, the repeated exposure of the mice to *S. mitis* EM-371 may have sensitised the lungs, leading to a stronger reaction by the immune system to the *NTi* infection. In addition, a possible biofilm formation in the lungs by the *S. mitis* may have favoured the colonization and persistence of the *NTHi*. Further studies are required to understand the synergic effect of the two strains in the exacerbation of the disease outcomes.

## 5. Conclusions

Overall, we have shown that interesting in silico and in vitro characteristics are not necessarily predictive clues for a beneficial effect in a preclinical model of lung infection. Indeed, a promising *S. mitis* strain, isolated from the buccal cavity of a human newborn, showed in vitro antimicrobial properties against respiratory pathogens but did not confer protection against an *NTHi* lung infection. Although we have not confirmed the protective effect of the *Streptococcus* strains in vivo, we propose that the model of *NTHi* infection we contributed to, described in detail, should be included as a critical go/no go step in the framework to screen potentially interesting probiotic strains for lung and respiratory applications.

## Figures and Tables

**Figure 1 nutrients-15-00263-f001:**
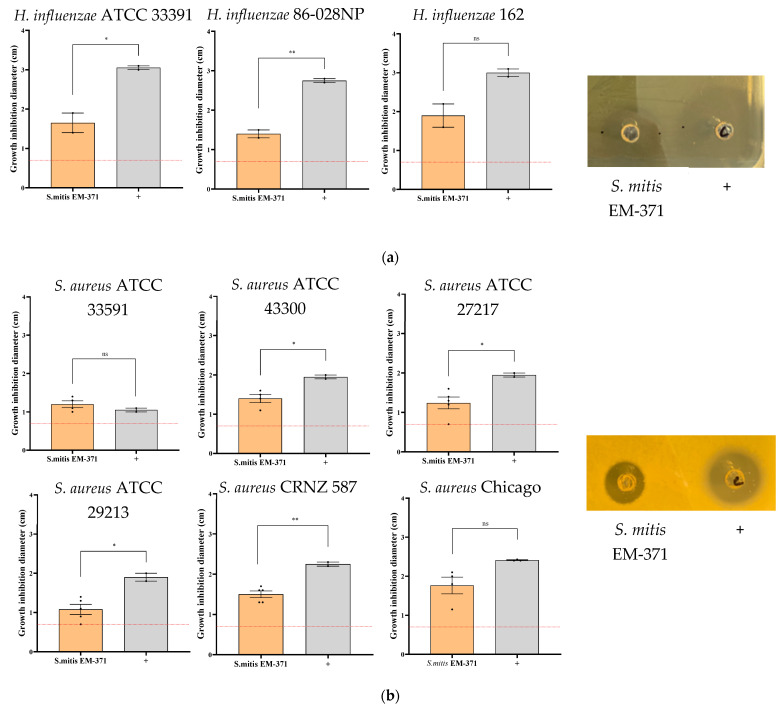
Growth inhibition zone induced by the *Streptococcus* EM-371 on several strains of (**a**) *Haemophilus influenzae*, (**b**) *Pseudomonas aeruginosa*, and (**c**) *Staphylococcus aureus*. Pictures are representative of at least 2 independent experiments. Graphics represent the mean of the growth inhibition diameter (in cm) in at least 2 experiments. Unpaired *t*-test was performed with GraphPad version 8 to determine the statistical significance. *p*-values: *: *p* < 0.03, **: *p* < 0.002. +: either chloramphenicol (against *H. influenzae* and *S. aureus* strains) or colistin (against *P. aeruginosa* strains). The red dotted line represents the diameter of the well.

**Figure 2 nutrients-15-00263-f002:**
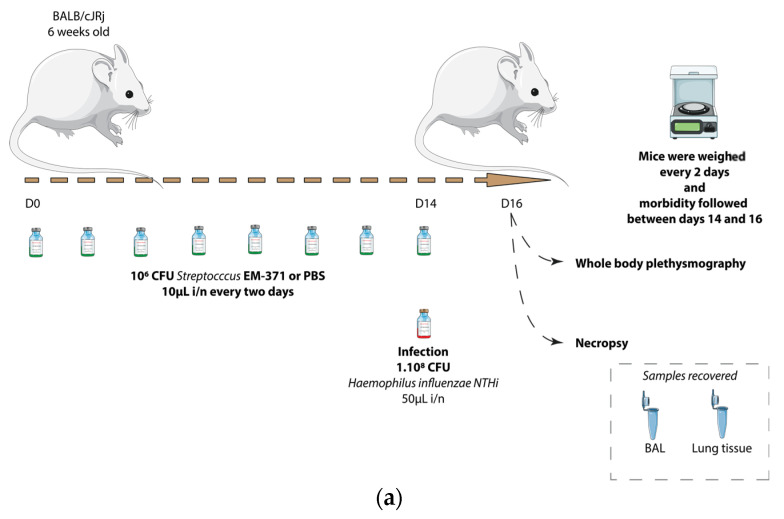
*Streptococcus* EM-371 did not protect against weight loss. (**a**) Experimental procedure to induce *Haemophilus influenzae* lung infection. (**b**) Mice were weighed daily. The growth curves are expressed as the mean ± s.e.m. of individual weights. To determine the statistical difference, the area under the curve was calculated and one-way ANOVA using Tukey’s multiple comparisons test was performed with GraphPad version 8. *p*-values: *: *p* < 0.03.

**Figure 3 nutrients-15-00263-f003:**
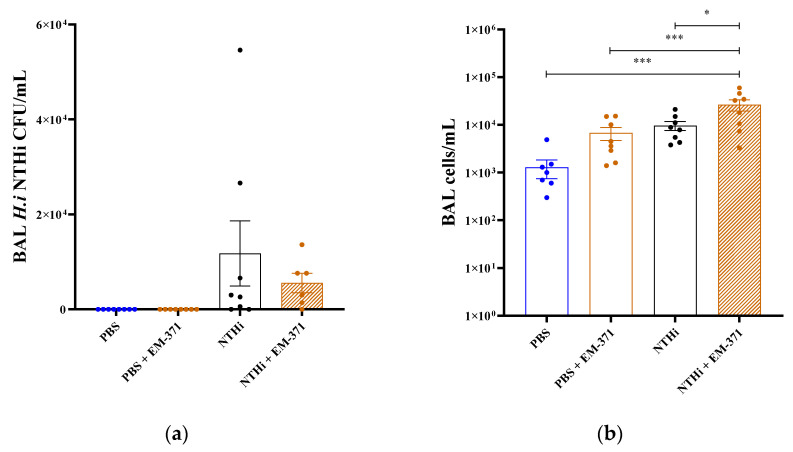
Infiltrating cells and viable *NTHi* recovered in BAL 48 h post infection. (**a**) BAL were diluted and plated on BHi agar supplemented with hemin and NAD. Colonies were enumerated after 48 h incubation at 37 °C. (**b**) BAL cells were enumerated using Malassez slides, cytocentrifuged and stained with May–Grünwald–Giemsa. The (**c**) neutrophils and (**d**) macrophages were enumerated and expressed as the number of cells/mL of BAL. One-way ANOVA using Tukey’s multiple comparisons test was performed with GraphPad version 8 to determine the statistical significance. *p*-values: *: *p* < 0.03, ***: *p* < 0.0002.

**Figure 4 nutrients-15-00263-f004:**
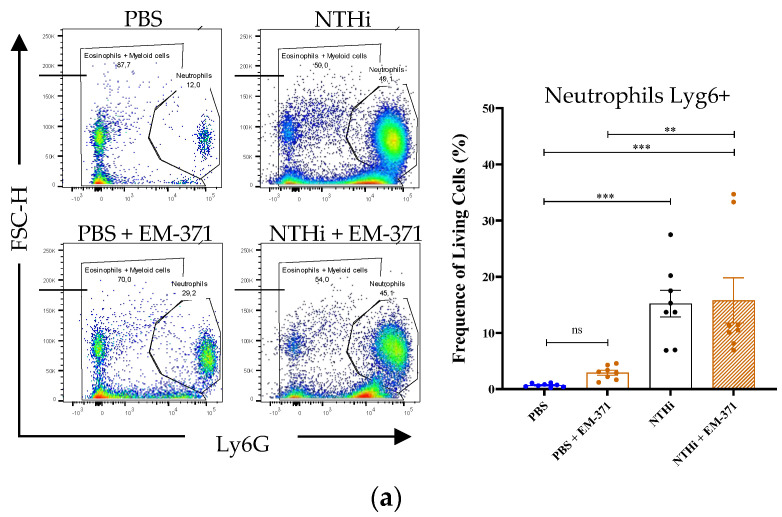
*NTHi* induced a prominent airway neutrophilia response. The frequency of (**a**) neutrophils Lyg6+, (**b**) DC CD11b+ and IM + Ly6C Mo, (**c**) CD11b- and AM were determined in lung tissue 48 h post-infection. The gating (**b**,**c**) are issued from the CD11b- and CD11b+ gatings, respectively (see Appendix A for the gating strategy). Values are expressed as the mean ± s.e.m. for one experiment (8 mice/group). To determine the statistical difference, one-way ANOVA using Tukey’s multiple comparisons test was performed with GraphPad version 8. *p*-values: *: *p* < 0.03, **: *p* < 0.002, ***: *p* < 0.0002 and ****: *p* < 0.0001.

**Figure 5 nutrients-15-00263-f005:**
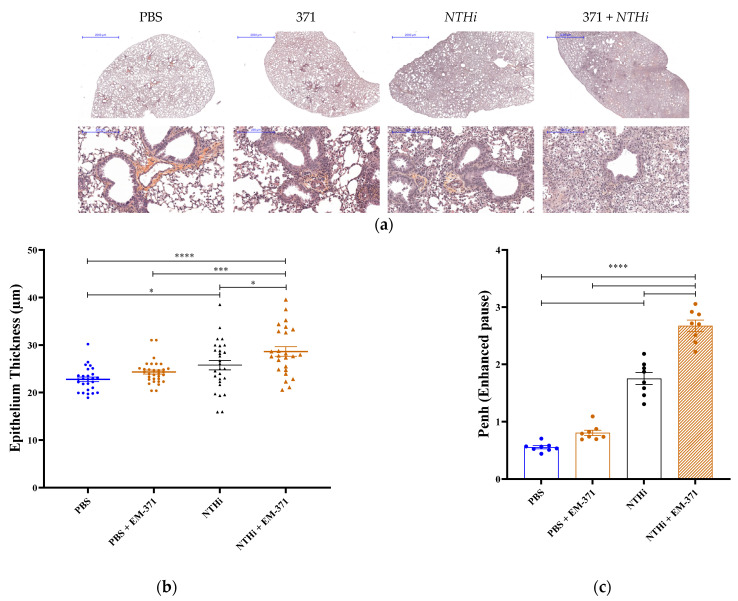
*NTHi*-challenged mice showed increased levels of inflammatory cell recruitment near the airway and epithelium thickening. (**a**) Lungs were fixed, embedded in paraffin and sectioned at a thickness of 5 μm. Lung sections were stained with hematoxylin–eosin–saffron (HES) and photographed using Case Viewer software. One representative lung section per group is shown. (**b**) Epithelium thickness was measured using Case Viewer software. The plots in the graph correspond to the measurements of four bronchi for at least six mice per group. Four representative bronchi were selected for each mouse and four measurements were performed per bronchus (up, down, left, right side). (**c**) Respiratory distress (Penh) measured by whole-body plethysmography. One-way ANOVA using Tukey’s multiple comparisons test was performed with GraphPad version 8 to determine the statistical significance. *p*-values: *: *p* < 0.03, ***: *p* < 0.0002 and ****: *p* < 0.0001.

**Table 1 nutrients-15-00263-t001:** Characteristics of *S. mitis* EM-371 that sustain the potential of this strain as a probiotic. * Data were obtained by whole genome analysis using RAST and PATRIC databases. ** Cytokine and chemokines data are from Mathieu et al., 2020. *** Amount of acetate quantified in the supernatant of an overnight bacterial culture.

Analysis	*S. mitis* EM-371
**Genome Analysis** Presence of genes potentially coding for adhesion sites and receptors (*)	Choline-binding protein: CbpD, lytC, pce Fibronectin-binding protein: pavA Plasminogen-binding protein: GAPDH Laminin-binding protein: lmb Other surface proteins: NanA, Eno, ZmpB, PsaA
**Genome Analysis** Presence of genes potentially associated to bacteriocins production (*)	blpT, blpZ, comE, blpR, pncP, tsaD, blpH, comA, cibA, pncG, comB, srtA, blpL and comD
**Modulation of** Cytokines/Chemokines— in comparison to untreated BEAS-2B cells (**)	Pro-inflammatory mediators not modulated: TNFα, IL-6 and IL-8 Mediators up-regulated: CCL25, Basic FGF, MIF
**Modulation of** Short-chain fatty acid (SCFA)— in culture supernatant (***)	Acetate (4 mM)

## Data Availability

The data presented in this study are available on request from the corresponding author.

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
