# Peer review of "An Isolate of Streptococcus mitis Displayed In Vitro Antimicrobial Activity and Deleterious Effect in a Preclinical Model of Lung Infection"

_nutrients, 2023, doi:10.3390/nu15020263_

Round 1

Reviewer 1 Report

In this study, Mathieu et al. investigated the antimicrobial and protective properties of the oral commensal S. mitis against clinically relevant pathogens in in vitro and in vivo settings. Although the study findings are interesting and have therapeutic implications, I have the following questions and concerns:

1.     Please mention the clear objectives of the study in the introduction.

2.     Were the mouse experiments repeated? If not, how did you ensure the reliability of the collected data?

3.     How were the frequency and dose of S. mitis instillations decided?

4.     I have concerns about the flow cytometric analysis:

a) Show the detailed gating strategy.

b) What controls were used? Isotype or FMO controls? 

c) Were the cell doublets excluded using FSC-H versus FSC-A plot?

d) Was live-dead cell gating (on flow cytometry) used to exclude dead cell?

e) In Figure 4, the labelling (e.g., PBS and NTHi) at the top of dot plots needs to be readjusted. Make it above the top line.

5. In the Figure 4, can you show the absolute number of cells?

6. Did you check if S. mitis was able to colonize the nasopharynx and oral cavity after nasal instillation?

7. Were the mice used in this study male, female or both?

8. Please elaborate a bit more on what could be the reasons behind exacerbated disease outcome in the S. mitis-treated mice.

9. Page number – 469: show the data as a supplementary information.

10. The protective efficacy of S. mitis was mainly ascertained in terms of weight loss and histopathology analysis. Were these findings corroborated by bacterial burden in the lungs and/or nasopharynx?

11. Please check the entire manuscript for errors/typos, such as ‘increase’ should be replaced by ‘increased’ (line 461) and ‘et’ should be replaced by ‘or’ (line 24). 

Author Response

Thank you for your constructive comments. Please find bellow our responses :

  1. Please mention the clear objectives of the study in the introduction: See multiple changes in the last paragraph of the introduction.
  2. Were the mouse experiments repeated? If not, how did you ensure the reliability of the collected data? : In our protocol, all tissular and functional parameters are coherent and show a deleterious effect of the strain. Moreover, we perform more than one experiment. For the first experiment (not published), we infected the mice with a lower volume of NTHi and we observed that the dose was not enough to infect the mice at the level of the published experiment. However, we observed that S.mitis EM-371 favoured lung infection when mice were exposed to a small volume of NTHi (we talk about this line 477). We also had a group infected with the same volume and concentration of NTHi than the published experiment. The results of body weight loss and NTHi load in BAL were similar between the two experiments. As said line 479, we have preliminary data showing that, a cocktail of two other bacterial strains showed neutral effect on the model (data not shown), the data are in accordance with the published data. These elements comfort us in the reliability of the collected data.
  3. How were the frequency and dose of S. mitis instillations decided? We reproduced the protocol (mode of administration) used and described by Remot et al (ISME J. 2017). The dose and the frequency were similar, because we are able to put in evidence both beneficial or deleterious effects in function of the strains considered.
  4. Concerns about the flow cytometric analysis:
    1. Show the detailed gating strategy. Please see Supplementary Figure S1
    2. What controls were used? Isotype or FMO controls? : Isotype control were used for every markers.
    3. Were the cell doublets excluded using FSC-H versus FSC-A plot? Yes they were. Information added line 204. Please see gating strategy in Supplemental Figure S2.
    4. Was live-dead cell gating (on flow cytometry) used to exclude dead cell? Yes it was. Using Zombie Aqua. Information added line 194. Please see gating strategy in Supplemental Figure S2.
    5. In Figure 4, the labelling (e.g., PBS and NTHi) at the top of dot plots needs to be readjusted. Make it above the top line.:  The labelling was replaced above the top line.
  5. In the Figure 4, can you show the absolute number of cells? : Sadly, a problem occurred with the count of the cells during the experiments. No data are available. Therefore, we are not able to translate the FACS analysis in absolute number of cells.
  6. Did you check if S. mitis was able to colonize the nasopharynx and oral cavity after nasal instillation? : No we didn’t check if S. mitis was able to colonize the nasopharynx and oral cavity after nasal instillation. However, a previous study by Remot et al (ISME J. 2017) showed that nasal administration enabled bacterial delivery to the lungs. Labeled strain were also found in the nasal cavity. It was verified by the detection of CFDA-SE labeled strains in BAL, lungs, nasal cavity and duodenum.
  7. Were the mice used in this study male, female or both? : The mice were males
  8. Please elaborate a bit more on what could be the reasons behind exacerbated disease outcome in the S. mitis-treated mice. : The following sentences were added to the discussion from line 483: “The mechanisms underlying the exacerbated effect of S. mitis were not deciphered in this study and remain unknown. However, the repeated exposition of the mice to S. mitis EM-371 may have sensitise the lungs leading to a stronger reaction of the immune system to NTHi infection. Also, possible biofilm formation in the lungs by S. mi-tis may have favoured the colonization and persistence of NTHi. Further studies are required to understand the synergic effect of the two strains in the exacerbation the disease outcomes.”
  9. Page number – 469: show the data as a supplementary information. Please see Supplemental Figure S2. Information has been added to line 476 (“Supplementary Figure S2”).
  10. The protective efficacy of S. mitis was mainly ascertained in terms of weight loss and histopathology analysis. Were these findings corroborated by bacterial burden in the lungs and/or nasopharynx? We did not determine the load of S. mitis in the lungs. Only the load of H.i NTHi was determined.

Reviewer 2 Report

In the manuscript by Matheiu et al, the potentially probiotic Streptococcus mitis strain E-371 described previously is further characterized. The authors find a moderate inhibitory effect of the EM-371 strain on H. influenzae and P. aeruginosa and minor effects on S. aureus upon direct contact in vitro. However, intranasal application of the strain in mice failed to prevent H. influenzae infection, but rather exacerbated the course of infection, including increased cellular infiltration of the lungs. The manuscript is clearly written and the experimental results support the colnclusions drawn by the authors. I suggest a careful revision of the manuscript in terms of proper English.

Minor point:

Line 32ff: The sentence starting with "Most of the bacterial..." implies that Saccharomyces is a bacterial genus. The sentence should be rephrased. 

Author Response

Thank you for the reviewing and your comments. The manuscript has been reviewed for errors/typos. The sentence line 32 has been rephrased.

Round 2

Reviewer 1 Report

The authors have revised the manuscript satisfactorily.